# Disease decreases variation in host community structure in an old-field grassland

**Rita L. Grunberg**[1]*, **Fletcher W. Halliday**[1¤a], **Robert W. Heckman**[1,2¤b], **Brooklynn N. Joyner**[1¤c], **Kayleigh R. O'Keeffe**[1¤d], **Charles E. Mitchell**[1,3]

1 Department of Biology, University of North Carolina, Chapel Hill, North Carolina, United States of America, 2 Department of Integrative Biology, University of Texas at Austin, Austin, Texas, United States of America, 3 Environment, Ecology and Energy Program, University of North Carolina, Chapel Hill, North Carolina, United States of America

¤a Current address: Department of Botany and Plant Pathology, Oregon State University, Corvallis, Oregon, United States of America
¤b Current address: USDA Forest Service, Rocky Mountain Research Station, Cedar City, Utah, United States of America
¤c Current address: Department of Parks, Recreation & Tourism Management, North Carolina State University, Raleigh, North Carolina, United States of America
¤d Current address: Department of Biological Sciences, Lehigh University, Bethlehem, Pennsylvania, United States of America
* rita.grunberg@duke.edu

**Data Availability Statement:** Data and code that support these findings are available available through Zenodo (10.5281/zenodo.6980274).

## Abstract

Disease may drive variation in host community structure by modifying the interplay of deterministic and stochastic processes that shape communities. For instance, deterministic processes like ecological selection can benefit species less impacted by disease. When communities have higher levels of disease and disease consistently selects for certain host species, this can reduce variation in host community composition. On the other hand, when host communities are less impacted by disease and selection is weaker, stochastic processes (e.g., drift, dispersal) may play a bigger role in host community structure, which can increase variation among communities. While effects of disease on host community structure have been quantified in field experiments, few have addressed the role of disease in modulating variation in structure among host communities. To address this, we conducted a field experiment spanning three years, using a tractable system: foliar fungal pathogens in an old-field grassland community dominated by the grass *Lolium arundinaceum*, tall fescue. We reduced foliar fungal disease burden in replicate host communities (experimental plots in intact vegetation) in three fungicide regimens that varied in the seasonal duration of fungicide treatment and included a fungicide-free control. We measured host diversity, biomass, and variation in community structure among replicate communities. Disease reduction generally decreased plant richness and increased aboveground biomass relative to communities experiencing ambient levels of disease. These changes in richness and aboveground biomass were consistent across years despite changes in structure of the plant communities over the experiment's three years. Importantly, disease reduction amplified host community variation, suggesting that disease diminished the degree to which host communities were structured by stochastic processes. These results of experimental disease reduction both

**Funding:** This work was supported by the NSF-USDA joint program in Ecology and Evolution of Infectious Diseases (USDA-NIFA AFRI grant 2016-67013-25762) awarded to CEM. FWH was supported by an Ambizione Grant (PZ00P3_202027) from the Swiss National Science Foundation. The funders had no role in study design, data collection and analysis, decision to publish, or preparation of the manuscript.

**Competing interests:** The authors have declared that no competing interests exist.

highlight the potential importance of stochastic processes in plant communities and reveal the potential for disease to regulate variation in host community structure.

## Introduction

Disease can be an agent of ecological selection (i.e., biotic and abiotic factors that contribute to species filtering, sensu Vellend 2010 [1]) and thereby influence the structure of ecological communities [2–5]. Field experiments reducing disease in plant communities have revealed that disease can not only increase local species richness, but also can select for certain host community compositions [6–9]. However, fewer studies have explicitly examined disease impacts on host community structure over time, i.e. host community dynamics [but see 6–8, 10, 11].

To understand how disease impacts host community dynamics, it is important to explore how disease relates to deterministic and stochastic processes that structure communities [1, 12, 13]. In host communities in which interspecific competition is an agent of ecological selection, disease can decrease the degree to which the community is dominated by better competitors and shift the relative abundance of some hosts [3, 5, 14, 15]. This could shape community structure by promoting communities to converge over time towards a similar composition of species less impacted by disease. We call this the convergence scenario. Importantly, the strength of convergence will also depend on the distribution and abundance of host species within each community [16]. Indeed, when host species vary in their response to disease and species composition varies among communities, communities may fail to converge [5, 17, 18]. Specifically, when a host species that is strongly impacted by disease is patchily distributed among communities, then the strength of selection by that disease will vary among communities, preventing convergence. We call this the spatial variation scenario. Further, communities may experience stochastic colonization and extinction of species that generate variation in community structure over time [1, 12, 19]. When stochastic processes are key drivers of community assembly, the colonization, extinction, and abundance of species could vary greatly among communities and drive differences in the impacts of disease over space and time, impeding convergence. We call this the stochastic variation scenario. Together, these three scenarios—convergence, spatial variation, and stochastic variation—emphasize that disease impacts on host communities will be related to their species composition and thus the processes that shape host communities over both time and space.

Disease, as an agent of ecological selection, can also interact with other processes that drive community assembly. Specifically, disease may reduce the relative influence of stochastic demographic processes, such as dispersal or ecological drift, on community structure [1, 20]. Over time or across multiple local communities, stochastic processes can increase variation in community structure [13, 21]. Thus, to the degree that disease reduces the relative influence of stochastic processes within host communities, disease may reduce variation in community structure over time and among communities, and this could promote community convergence. On the other hand, if host community composition is relatively homogenous, disease may not be related to variation among host communities. As such, disease could have variable impacts on host community structure, depending on the relative importance of disease, along with other agents of ecological selection, and stochastic processes that structure communities. The relationship between disease and variability in host community composition is important as it can set the stage for long-term differences in host community assembly.

The role of disease as an agent of ecological selection will also depend on the structure of the parasite community. Hosts are commonly infected with a diverse community of parasite

species that vary in their degree of host specificity and their effects on host fitness [22]. In some systems, generalist parasites can shift competitive interactions among hosts (i.e., apparent competition) and specialist parasites also generate fitness differences between hosts [5]. Thus, variation in parasite community structure can drive disease impacts on host individuals [23] and communities [10, 24]. Moreover, parasite infections are often seasonal, so disease burdens and parasite community structure will vary over time [25]. Previous field experiments have found that parasite impacts on the host community can depend on parasite community composition [10, 24] and that parasite community composition can shift seasonally [25–27]. But, the effect of seasonal variation in parasite community composition on plant community structure remains untested.

Here, we experimentally manipulated seasonal burdens of foliar fungal disease, and aimed to alter parasite community structure, by treating intact plant communities with fungicide for different periods of the growing season. In our experiment, we quantified not only effects of disease on the species richness, composition, and biomass of local plant communities, but also impacts of disease on variation among communities over three years. Quantifying variation among communities over time is important because consistent effects of disease reduction on plant species richness within communities may not translate to consistent shifts in plant community composition (i.e. community convergence) across replicate communities within each fungicide treatment group.

Our experiment focused on disease impacts within a long-established grassland community dominated by tall fescue. Given the dominance of tall fescue in this system, disease impacts on this host species may contribute to differences in plant species richness and community composition. Convergence in plant community composition in response to experimental disease reduction could depend on the compositional variation present among communities. On one hand, the convergence scenario predicts that the species that vary in presence or abundance among experimental communities may not be species that are strongly impacted by disease reduction. In our system, grass species appear to make a tradeoff between growth and defense [28]. If this tradeoff determines plant community structure, then experimentally reducing disease could strongly release fast-growing species, allowing them to rise in dominance and leading to convergence in community structure. On the other hand, the spatial variation scenario predicts that community convergence could be prevented by standing variation among communities in species composition, particularly species other than tall fescue. Further, the stochastic variation scenario predicts that community convergence could be impeded if stochastic processes during the experiment generate high variation in species composition among experimental communities. Our study examined these three scenarios by analyzing effects of disease reduction on variation among communities over time.

## Methods

### Experimental design

Our experiment was conducted at Widener Farm in the Duke Forest Teaching and Research Laboratory in Orange Co., North Carolina, USA. We were given permission to conduct this experiment by Duke Forest under research registration number: R1617-434. This old field site was previously used to produce row crops until 1996. Since then, the site has been mowed at least once a year in the summer to maintain herbaceous plant dominance. In our plots, the dominant plant species was tall fescue, *Lolium arundinaceum* (= *Festuca arundinacea*) (S1 Fig), which supports both a heavy load of foliar fungal disease and a great diversity of parasites [29]. Other abundant species included *Lespedeza cuneata*, *Lonicera japonica*, *Sorghum halepense*, *Verbesina occidentalis* and *Tripsacum dactyloides* (S1 Fig).

We used the observed natural epidemic of foliar fungal disease on the most abundant host, tall fescue, to design our fungicide treatments that aim to reduce disease and shift the composition of parasites causing disease in our system. Epidemics of foliar fungal disease on tall fescue are seasonal; typically, the disease anthracnose (caused chiefly although not exclusively by *Colletotrichum cereale)* peaks in late spring, while brown patch (caused by *Rhizoctonia solani)* peaks in the summer and crown rust (caused by *Puccinia coronata)* peaks in the early fall (Fig 1). Hereafter, we refer to these diseases rather than the causal parasite species because our

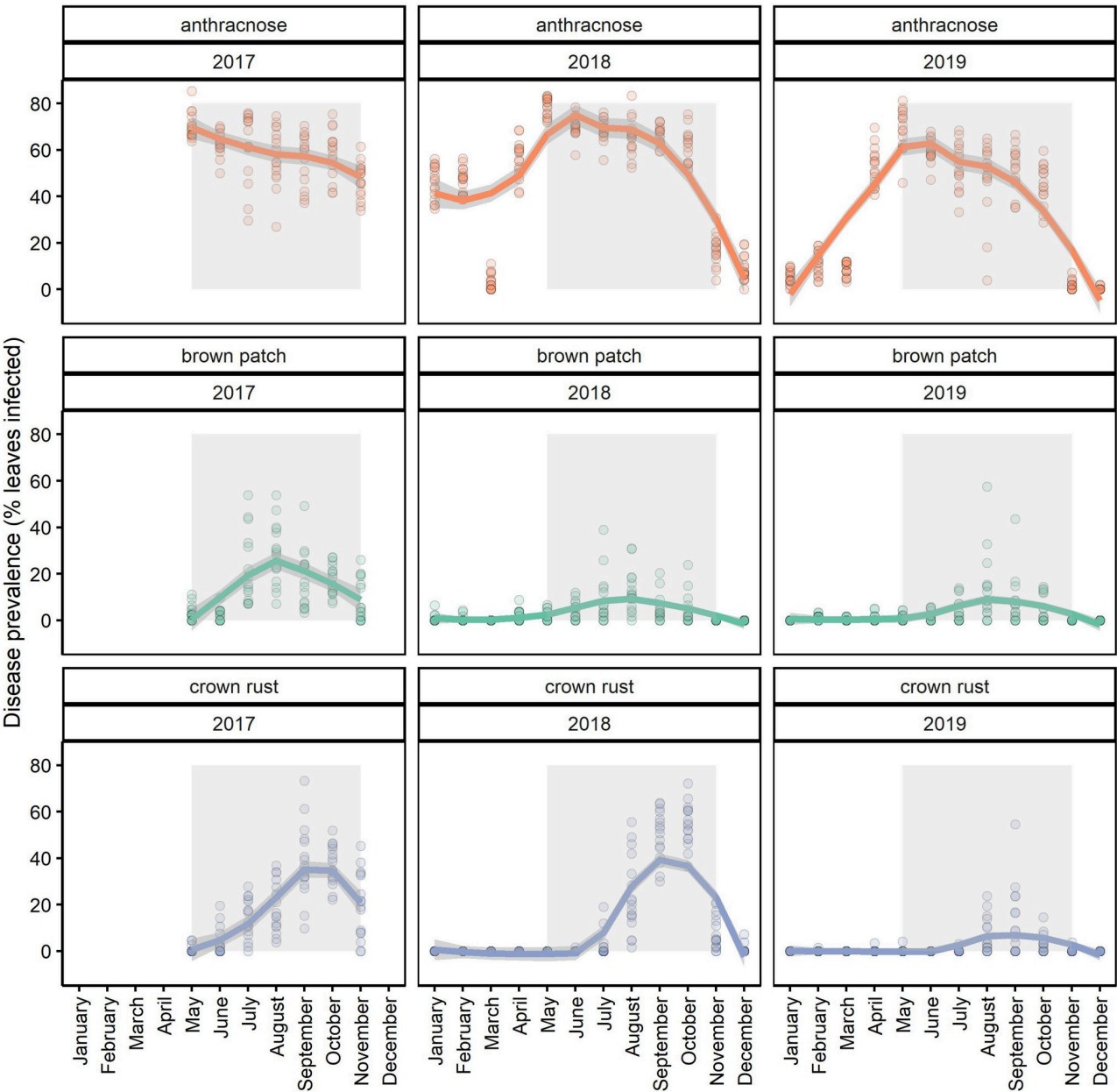

**Fig 1. Seasonal disease epidemics on leaves of tall fescue from control (never sprayed with fungicide) plots.** Generally, anthracnose peaks in late spring, while brown patch peaks in summer and crown rust in fall. The grey rectangle delineates the disease survey data used to calculate disease burden in analyses (i.e., the area under the disease progress stairs (AUDPS) of disease prevalence from May-November). Loess fitted lines are drawn to show the peak prevalence of each disease, and points are the raw disease prevalence data for each of the 16 control plots.

study focuses on disease impacts. These diseases can have a wide host range, infecting other plant species in this system [29], although it is possible that the same disease symptom on different plant species is caused by cryptic species of fungi [30]. Anthracnose and brown patch are caused by hemibiotrophic and necrotrophic parasites, respectively, and have been observed to infect a variety of host species in this system. However, crown rust is an obligate biotroph that tends to be more host specific but has been observed to infect other hosts in this system. In addition, the transmission biology of these diseases differ in key ways; anthracnose is spread by dispersal of conidia via water splash, brown patch is soil-borne, and crown rust is transmitted through airborne spores.

Our fungicide treatment regimens correspond with the peak of the seasonal epidemics of these diseases, so that different fungicide treatments would shift the composition of disease. Treatments varied in their duration of fungicide exposure including: (1) no fungicide (control) to represent ambient levels of all diseases, (2) fungicide until mid-July (approximately 7 months, soon after the typical start of brown patch epidemics) to reduce the seasonal window of anthracnose infections, (3) fungicide until mid-September (approximately nine months, soon after the typical start of crown rust epidemics) to reduce both the anthracnose and brown patch seasonal infection window, and (4) year-round application of fungicide to reduce all diseases. Fungicide application started each year in January, except in 2017, when fungicide treatments started in May. Throughout, we refer to these fungicide treatments with respect to their designated yearly duration of fungicide application: never sprayed, seven months, nine months, and year-round. Fungicide treatments started in May 2017 and ended in February 2020.

The non-systemic fungicide *Dithane75DF Rainshield*, (75% mancozeb, Dow AgroSciences, Indianapolis, Indiana, USA) was sprayed onto foliage at the recommended rate (20 g fungicide/1 gal of water) to reduce foliar fungal disease; the volume applied was adjusted as vegetation grew and senesced to cover all leaves with the fungicide. Prior studies have found that application of this fungicide at recommended rates did not affect plant growth or mycorrhizal colonization [31], including growth of tall fescue and other common species from our experimental site [7]. Fungicide was applied approximately every two weeks during the designated fungicide treatment time. Plots not assigned to receive fungicide treatments were instead sprayed with water. In 2018, plots were washed with water at the end of each seasonal treatment's fungicide application period (i.e., in mid-July for the seven-month treatment and mid-September for the nine-month treatment) to remove fungicide from leaves and allow fungal colonization of the plots for the rest of the year. Plots in all treatments were washed for the same duration to equalize water addition among plots. Plots were mown annually at the end of each growing season to reduce the establishment of woody vegetation, which tends to outcompete herbaceous species in Eastern North American old-fields, including our site [32, 33].

In total, we established 64 experimental plots in an intact old-field that was fenced to exclude vertebrates, particularly deer and rodents (Fig 2). At the beginning of the experiment on 11–13 June 2017, we increased tall fescue dominance of plant communities by clipping shoots of two other abundant species, *Lespedeza cuneata* and *Sorghum halepense*, at their base. Experimental plots were assigned to one of four fungicide treatments in a fully randomized design. Each plot was 2 m × 2 m and separated by 1 m with a 2-m mowed buffer surrounding the entire experimental area, and the 64 plots were arranged in a 16×4-plot array with a total footprint of 51 m × 15 m.

## Data collection

Plant community composition was recorded on 09 November 2017, 10 October 2018, and 10 October 2019 by R. Heckman. The cover of all plant species, including litter and bare ground,

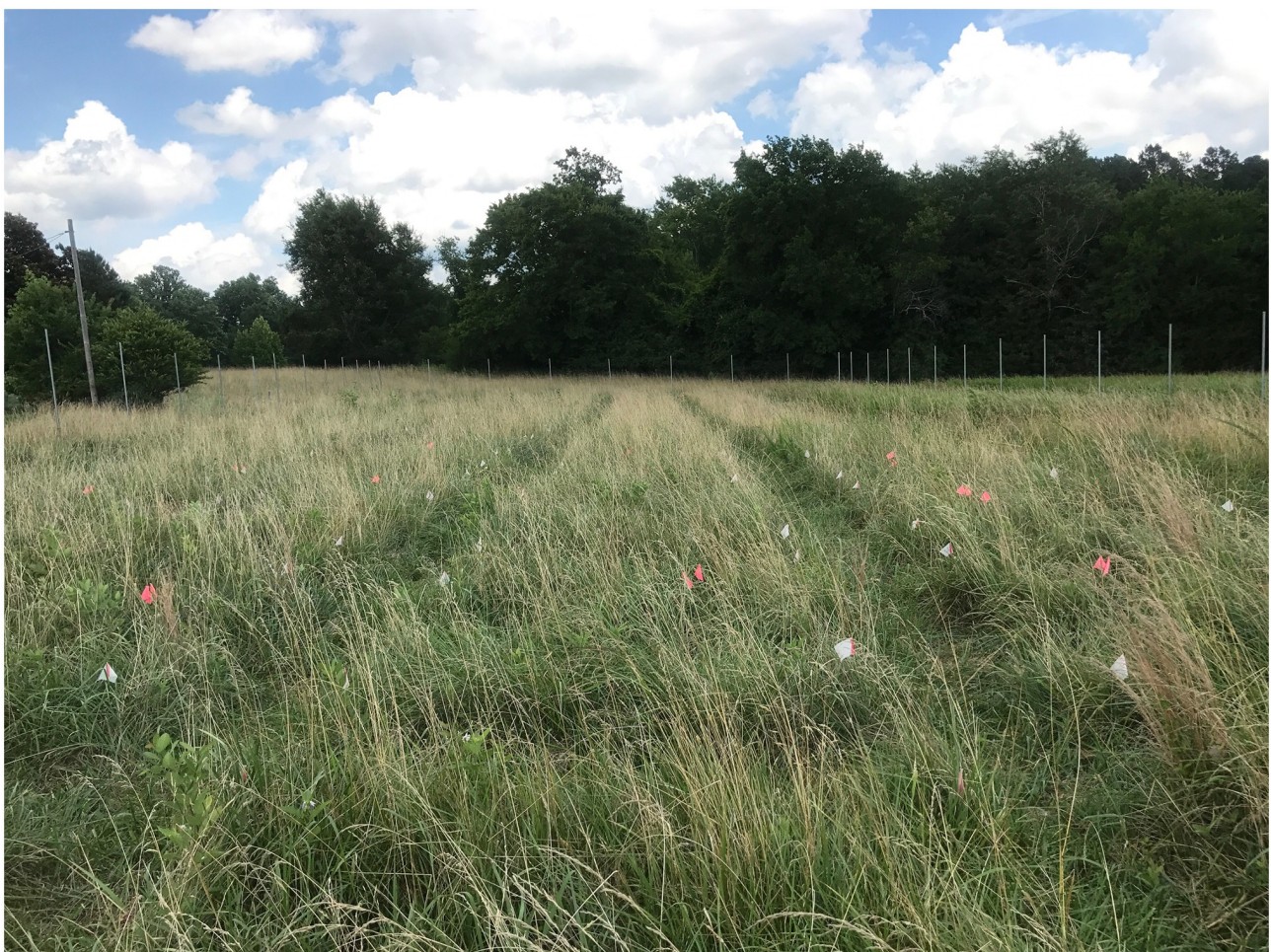

**Fig 2. Photograph taken at the start of the experiment in the summer of 2017.** This shows a subset of the experimental plots established in an old-field grassland community dominated by tall fescue. Photo taken by B. N. Joyner.

was visually estimated within a 0.75 m × 0.75 m permanent quadrat in each plot. The absolute cover of each plant species was estimated independently as plants may overlap, and thus total cover often exceeded 100% in plots. We then calculated relative cover for each species in a plot as the absolute cover of that species divided by the sum of absolute cover of all species in a plot. Litter and bare ground were not included in any statistical analysis of plant communities.

We evaluated the effectiveness of the fungicide treatments by surveying disease prevalence in tall fescue. Disease prevalence was not assayed for other plant species; we rationalized that quantifying disease on the most abundant host species, tall fescue, which initially constituted ~87% of plant cover in our plots (S1B Fig), provides a useful estimate for the overall effectiveness of the fungicide on most diseases in this system. We surveyed disease monthly, starting in March 2017 and ending in December 2019, by haphazardly selecting 20 tillers (i.e., individual grass shoots) of tall fescue within each plot and recording visible disease symptoms on all leaves on a tiller. We then quantified disease prevalence within a plot as the total number of infected leaves (infected = any amount of disease) divided by all leaves surveyed across the 20 plants, treating leaves as host individuals because fungal infections are localized within a leaf [26].

We measured aboveground plant biomass each year in mid-November by harvesting the entire 0.75 m × 0.75 m quadrat in 2017, and by harvesting two 0.5 m × 0.2 m strips of vegetation in 2018 and 2019. Dead vegetation from the growing season (litter), which did not include prior year's biomass, was included in biomass measurements. In 2018, we sorted plant biomass into five categories: tall fescue, non-fescue monocots, non-woody dicots, woody dicots, and litter. Aboveground biomass was then oven-dried at 65 ˚C for at least 72 hours and weighed.

## Analysis

**Fungicide treatment effects on disease burden and composition.** We quantified host population-level disease burden by calculating the annual area under the disease progress stairs (AUDPS) [34] using the monthly disease prevalence survey data. This measure of disease burden (i.e., AUDPS) provides an advantage over other measures like the area under the disease progress curve because it gives a better estimate of the contribution of the first and last observation [34]. To calculate disease burden, we used disease survey data from May until November of each year to be consistent across years; also, this timespan represents the bulk of the growing season in our system. The AUDPS, our measurement of disease burden, was estimated using the 'agricolae' R package [35]. We calculated a cross-disease disease burden as the proportion of tall fescue leaves infected by any of the diseases (anthracnose, brown patch, and crown rust) as an indicator of overall disease burden.

We also include disease burden data for each of the three diseases infecting tall fescue. As a measure of parasite composition, we also quantified the relative contribution of each disease's AUDPS to the cross-species AUDPS. Quantifying the relative contribution of each disease allows us to account for overall differences in disease due to fungicide. We assessed the impact of the fungicide treatment over time on each disease's relative contribution to the cross-species AUDPS using a multivariate generalized linear model with the 'mvabund' R package to test for changes in parasite composition.

The effects of the fungicide treatments on disease burden over time were evaluated using a repeated measures analysis of variance (ANOVA). In this analysis, we used the annual cross-disease burden (i.e., AUPDS). In the ANOVA we included fungicide treatment, year, and treatment*year interaction as fixed effects, and experimental plot as a random effect. As a measure of disease reduction, we also report the disease burden log response ratio relative to the control-no fungicide plots.

**Plant richness, Hill-diversity, and biomass.** We used repeated-measures ANOVAs to test the effects of fungicide treatment over time on plant biomass and plant diversity metrics. Plant diversity was quantified using three metrics based on Hill's series of diversity [36] in the 'vegan' package [37]. The value of $q$ in Hill's series is related to differences in the weighting of the relative abundance of taxa: taxonomic richness ($q = 0$, no abundance weighting), Hill-Shannon diversity ($q = 1$, provides a balanced measure of both rare and common species), and Hill-Simpson diversity ($q = 2$, emphasizes common species) [38]. All ANOVAs included fungicide treatment, year, and fungicide treatment*year interaction as fixed effects and experimental plot as a random effect to account for repeated measures.

Significance tests were based on type 3 tests in the 'afex' package [39]. As a measure of the effect size for each variable in a model, we report $\eta^2_{partial}$, which is calculated as the ratio of the variance explained by the variable to the sum of that explained variance plus the residual error variance. For each variable in a model, $\eta^2_{partial}$ is bound between 0 to 1. We then evaluated differences among fungicide treatments and between years using Tukey's Honestly Significant Difference (HSD) post-hoc tests. We checked assumptions of normality and homogeneity of

variance using diagnostic plots of model residuals. To minimize heteroscedasticity, we log-transformed plant richness and biomass.

**Plant community composition.** We assessed whether fungicide treatments influenced plant community composition over time by modelling the relative cover of plant species using a multivariate generalized linear model in the 'mvabund' R package with a negative binomial distribution [40]. This analysis allows us to detect both community-level and species-level responses to fungicide treatments. When considering species-level responses, we used univariate tests that were adjusted for multiple comparisons through resampling based on the Holm step-down procedure [40]. We accounted for repeated sampling of communities by restricting permutations within blocks that correspond to the identity of the experimental plot using the 'bootID' argument ($n$ = 999 permutations). Patterns of plant community dissimilarity were then visualized using nonmetric multidimensional scaling based on Bray-Curtis distances of plant relative cover in R package 'vegan' [37].

We expected the colonization and loss of plant species in our experimental plots to contribute to variation in community structure among plots over time, and we predicted that fungicide treatments would interact with this variation. Specifically, disease reduction could further increase variability in community structure by reducing selection via disease and thus increase within-treatment variation. Therefore, we also examined whether within-treatment variation in community structure, measured as the distance from the Bray-Curtis treatment centroid of each year, differed among treatments and years using a repeated measures ANOVA. We used experimental plot as a random effect to account for repeated measures, and log-transformed distance from the centroid to account for heteroscedasticity. All data and code are publicly available through zenodo at DOI: 10.5281/zenodo.6980274.

# Results

## Fungicide treatment effects on disease burden and composition

The application of fungicide decreased cumulative foliar fungal disease burden over the growing season, as measured across the three common diseases of tall fescue ($F_{3,60}$ = 176.63, $p < 0.001$, $\eta^2_{partial}$ = 0.89, S1 Table, S2 and S3 Figs). In plots that received year-round fungicide, disease was reduced by 42.4% in 2017, 35.6% in 2018, and 85.9% in 2019 relative to the control (S1 Table). In addition, disease burdens declined from 2018 to 2019 by on average 58% ($F_{2,120}$ = 1610.82, p < 0.001, $\eta^2_{partial}$ = 0.96), and the effects of fungicide treatment on disease varied among years (treatment*year: $F_{6,120}$ = 51.76, $p < 0.0001$, $\eta^2_{partial}$ = 0.72, S2 Fig). In the first year of the experiment (2017), disease burdens from plots experiencing year-round fungicide were significantly lower than those from plots sprayed for nine months (year-round v. nine-month contrast for 2017, $p < 0.001$). In subsequent years, disease burdens between those fungicide treatment groups were not statistically distinguishable from each other (year-round v. nine-month 2018 contrast, $p$ = 0.37; 2019 contrast, $p$ = 0.98, S2 Fig). Despite that decrease in statistical significance and the large variation among years, overall, the treatments that applied fungicide over a greater fraction of the growing season tended to reduce disease burden more over the growing season.

Generally, anthracnose accounted for most of the disease in tall fescue throughout the experiment and had the highest relative contribution to the cross-species AUDPS across all fungicide treatments and years (mean relative contribution = 0.93, range: 0.53 to 1.0). Rust (mean, = 0.11, range: 0.0 to 0.42) and brown patch (mean = 0.07, range: 0.0 to 0.39) infections were less frequent overall (S4 Fig). The community composition of foliar diseases remained relatively consistent across treatments (multivariate glm: p = 0.246) but did vary over time (p = 0.008). There was no interaction between fungicide and time (p = 0.995). Univariate tests

conducted on each disease indicate an impact of the fungicide treatment on rust infections (univariate test, p = 0.001). Specifically, the relative contribution of rust was lower in the 9 month and year-round fungicide treated plots. Thus, although all fungicide treatments reduced disease, seasonal fungicide application did not alter parasite community composition.

## Plant richness and Hill-diversity

Fungicide treatments generally reduced plant diversity, but with important variation among diversity metrics. Plant richness varied among years (year: $F_{2,120} = 72.18$, $p < 0.001$, $\eta^2_{partial} = 0.42$), and was reduced by fungicide treatment (treatment: $F_{3,60} = 3.81$, $p = 0.014$, $\eta^2_{partial} = 0.07$). Plant richness was comparable in 2017 (mean richness = 4.59) and 2019 (mean richness = 4.14; Tukey HSD $p = 0.089$), but relatively higher in 2018 (mean richness = 7.16; Tukey HSD $p < 0.001$, Fig 3, left column of panels). Still, the effect of fungicide treatment on plant richness did not change between years (treatment * year: $F_{6,120} = 1.20$, $p = 0.32$). Control plots that were never sprayed with fungicide, experiencing ambient levels of disease, on average, had 1.2 more plant species than plots treated with fungicide (Fig 3, left column of panels). Plant richness did not differ between the three fungicide-treated groups (Tukey HSD $p > 0.05$). In summary, fungicide reduced plant richness independently of the duration of fungicide exposure, and the effect of fungicide on richness was consistent across years despite changes in plant richness.

Fungicide treatment reduced plant Hill-Shannon diversity ($F_{3,60} = 5.42$, $p = 0.002$, $\eta^2_{partial} = 0.12$) and Hill-Simpson diversity, although this latter main effect was only marginally significant ($F_{3,60} = 2.68$, $p = 0.06$, $\eta^2_{partial} = 0.07$). Between-year variation in both Hill-Shannon and Hill-Simpson diversity was large and a main driver of plant diversity in this experiment (year: Shannon $F_{2,120} = 154.44$, $p < 0.001$, $\eta^2_{partial} = 0.55$; Simpson $F_{2,120} = 100.85$, $p < 0.001$, $\eta^2_{partial} = 0.44$, Fig 3). Moreover, there was a significant interaction between fungicide treatment and time for both diversity measures (treatment*year: Shannon $F_{6,120} = 5.49$, $p \leq 0.001$, $\eta^2_{partial} = 0.12$; Simpson $F_{6,120} = 4.19$, $p = 0.001$, $\eta^2_{partial} = 0.09$). Hill-Shannon and Hill-Simpson diversity did not differ among treatment groups in 2017, the first year of experiment (Tukey HSD $p > 0.05$). In 2018, both Hill diversity metrics were higher in the control plots than the fungicide-treated plots, except for Hill-Simpson diversity in plots treated with fungicide year-round (Tukey HSD: Fig 3). Finally, in 2019 only the control and year-round fungicide-treated plots differed in plant diversity (Tukey HSD: Shannon $p = 0.014$, Simpson $p = 0.017$). In contrast to plant richness, the effects of fungicide on Hill-Shannon and Hill-Simpson diversity were more variable among fungicide treatment groups, and only became evident after the first year of the experiment.

## Plant biomass

Plant community biomass increased in response to the fungicide treatments (treatment: $F_{3,60} = 20.27$, $p < 0.001$, $\eta^2_{partial} = 0.25$) (Fig 4). Year-round fungicide treatment increased plot biomass by 14.9% in 2017, 47.7% in 2018, and 46.6% in 2019 relative to the control (i.e., no fungicide) plots. Similar increases in plant biomass also occurred in the plots treated with fungicide for nine months (% change relative to control: 2017 = 12.1%, 2018 = 44.4%, 2019 = 45.3%). Increases in plant biomass from plots exposed to fungicide for seven months were about half as large as those from the other fungicide treatments (percent change relative to control: 2017 = 5.1%, 2018 = 26.7%, 2019 = 26.5%). Similar fungicide treatment effects were observed in the 2018 sorted biomass data, and differences in 2018 plant biomass were chiefly driven by an increase in fescue biomass within fungicide-treated plots (S5 Fig). Across all treatments, plant biomass increased over time and was highest in 2019 (year: $F_{2,120} = 64.14$, $p < 0.001$,

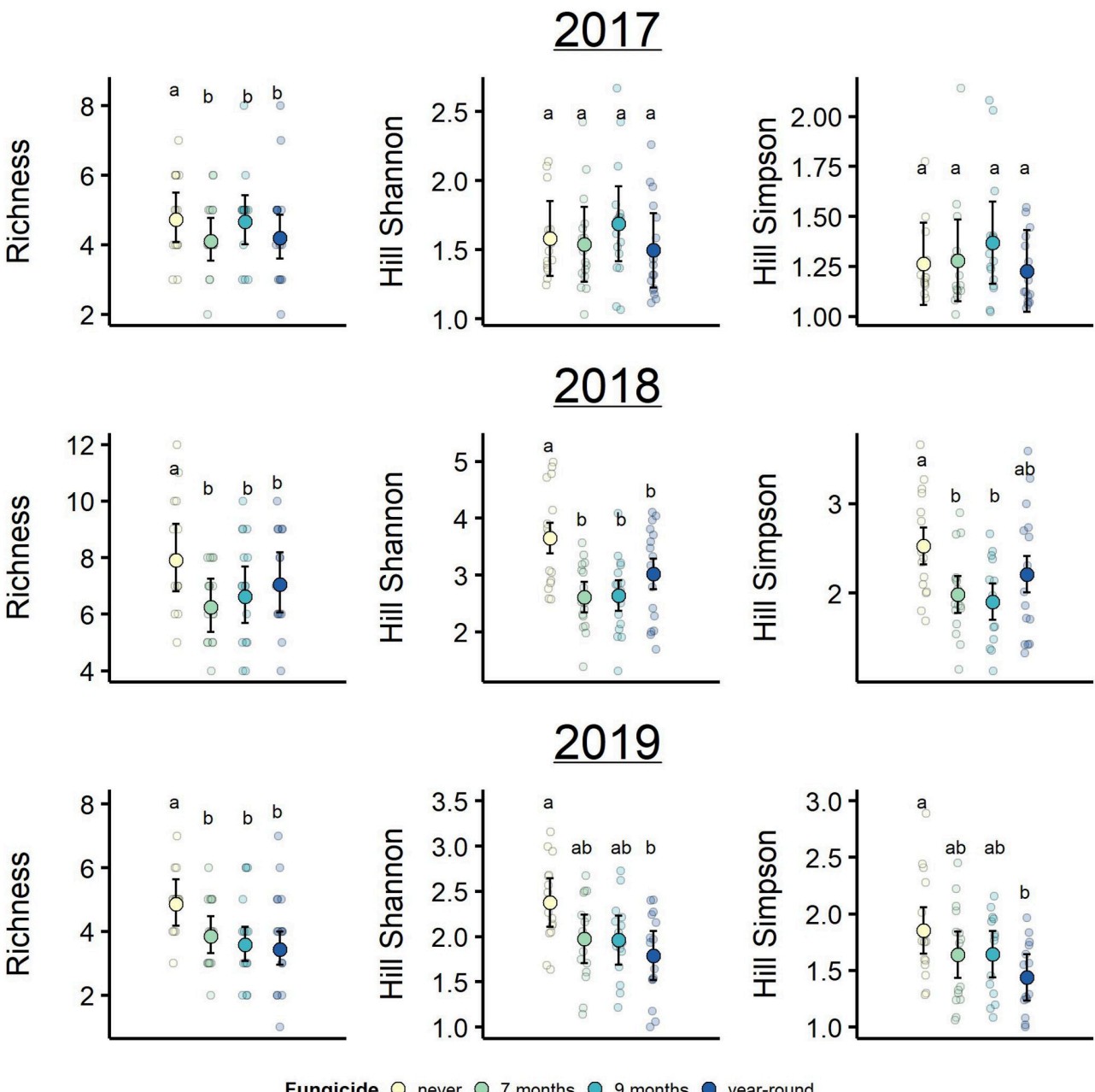

**Fig 3. Plant diversity, measured as taxonomic richness, Hill-Shannon diversity, and Hill-Simpson diversity was generally reduced by fungicide treatments and varied across years.** Means not sharing the same lower-case letters indicate significant differences between fungicide treatment groups. Due to the treatment*time interaction, Hill Shannon and Simpson diversity groupings are based only on comparisons between treatments within each year. Plotted are observed treatment means and their associated 95% confidence intervals, and smaller points display the raw data, which were jittered for visualization.

$\eta^2{}_{partial} = 0.42$). There was weak evidence of an interaction between fungicide treatment and time on plant biomass, which may have resulted from an increase in fungicide effects on biomass in 2018 and 2019 (treatment*year: $F_{6,120} = 1.99$, $p = 0.07$). Overall, differences in plant community biomass were related to variation in fungicide exposure throughout the growing season, so that communities exposed to fungicide for a longer duration had greater biomass.

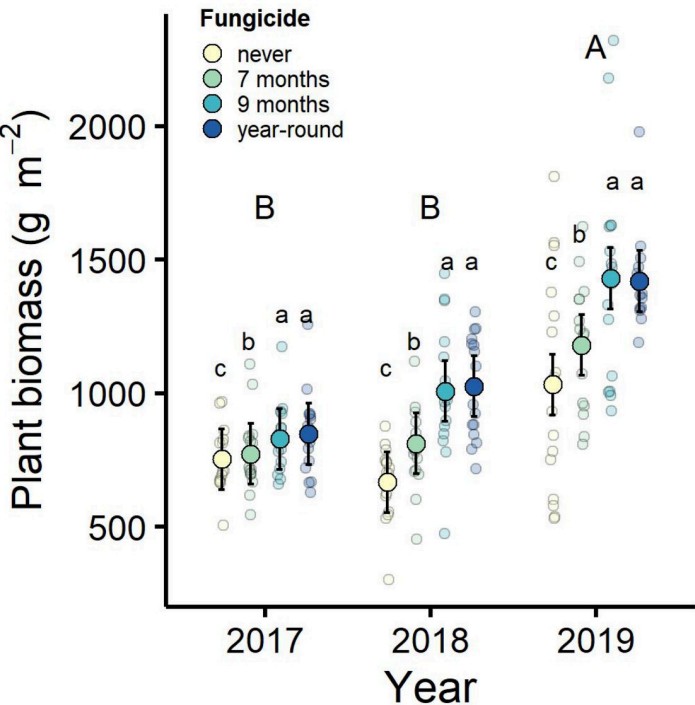

**Fig 4. Plant community biomass generally increased in response to fungicide treatments.** Years not sharing the same capital letters indicate significant differences between years based on Tukey's HSD post-hoc tests. Treatments not sharing the same lower-case letters indicate significant differences among fungicide treatment groups. Plotted are observed treatment means and their associated 95% confidence intervals, and smaller points display the raw data, which were jittered for visualization.

## Plant community composition

Plant communities did not converge towards a similar community composition. The species-level composition of plant communities changed over time (multivariate glm: deviance = 34.73, $p < 0.001$), but changes in plant community composition was not related to the fungicide treatments (treatment: deviance = 26.46, $p = 0.897$, S6 and S7 Figs). Additionally, the temporal changes in plant community composition did not interact with the fungicide treatments (treatment *time: deviance = 4.85, $p = 0.718$). For example, the relative cover of the dominant plant, tall fescue, was not affected by fungicide treatments ($p = 0.45$), but generally declined over time (univariate test, $p > 0.001$). In the first year of the experiment in 2017, all plots were dominated by tall fescue (mean relative % cover = 87.48, sd = 8.59), then fescue declined in 2018 (mean relative % cover = 62.37, sd = 16.31) and 2019 (mean relative % cover = 69.49, sd = 20.30). Similarly, tall fescue absolute percent cover was not impacted by fungicide treatments and declined over time (mean absolute % cover: 2017 = 95.9, 2018 = 83.0, 2019 = 82.4), so these results were not driven by differences in total absolute percent cover of all species. We could not identify any plant species that differed in relative abundance between fungicide treatments over time (all univariate species-level tests: $p$-adjusted $> 0.05$). Ultimately, the fungicide treatments did not affect species-level composition of plant communities, indicating communities were not converging towards a similar community state under lower disease.

Although fungicide treatments did not affect species-level composition of plant communities, the fungicide treatments tended to increase variation in plant community composition (treatment: $F_{3,60} = 11.05$, $p < 0.001$ $\eta^2_{partial} = 0.260$). Plant community composition varied

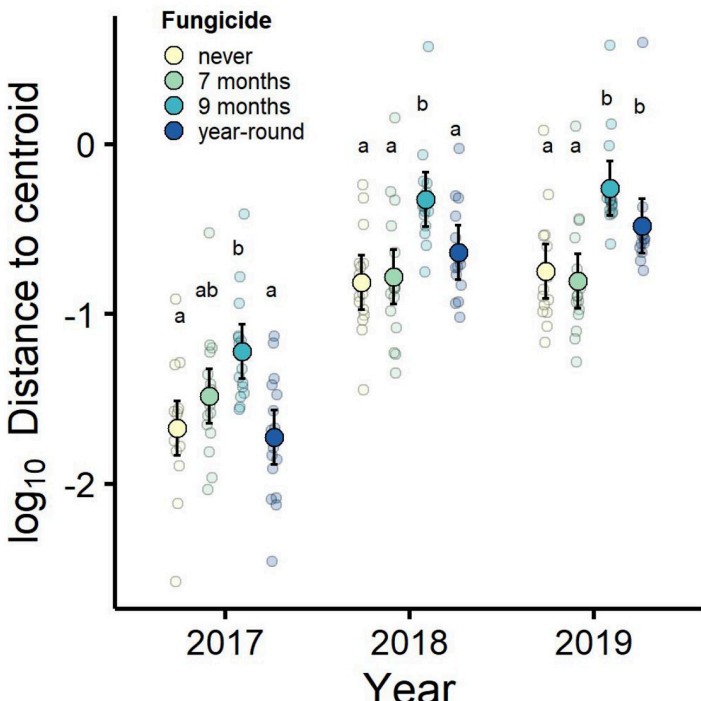

**Fig 5. Variation in plant community composition increased over time and with seasonal duration of fungicide treatment.** The fungicide treatments tended to increase variation in plant community composition, and that increase in variation tended to amplify over time, particularly in plant communities treated with fungicide for nine months out of the year or year-round. Within-treatment variation in plant community composition was measured within years as the log-transformed distance to the Bray-Curtis community centroid. Plotted are observed treatment means on log-transformed data and their associated 95% confidence intervals, and smaller points display all data, which were jittered for visualization. Treatment means not sharing the same letters denote significant differences based on Tukey's post-hoc comparisons of fungicide treatments within each year.

more among plots within the year-round fungicide treatment and more among plots within the nine-month fungicide treatment than within the seven-month and never-sprayed fungicide treatment groups (Tukey HSD $p < 0.05$, Fig 5). In addition, variation in plant community composition among plots increased over time (year: $F_{2,120} = 307.01$, $p < 0.001$ $\eta^2_{partial} = 0.651$), and the effect of fungicide treatments on variation in plant community composition changed over time (treatment*year $F_{6,120} = 3.55$, $p = 0.004$, $\eta^2_{partial} = 0.061$). This interaction was driven by greater increases over time in variation among the plots sprayed with fungicide year-round (Tukey HSD, Fig 5). In contrast, plots sprayed with fungicide for 9 months of the year had consistently higher variation in host community structure and did not display differences in treatment effects over time. This indicates that these disease-reduction treatments amplified variation in host community structure over time. While fungicide treatments did not explain differences in the average plant community composition, variation in community composition within fungicide treatment groups was greater in plots treated with fungicide for a longer time each growing season.

## Discussion

Experimentally reducing foliar fungal disease in intact plant communities decreased plant diversity and increased plant biomass in our experiment. However, disease reduction did not

lead to consistent shifts in the relative abundance of plant species and more generally, communities within a given disease-reduction treatment group did not converge towards a similar composition. Instead, plant community composition varied considerably among experimental plots and over time, and this among-plot variation in plant composition was amplified by disease reduction, suggesting that under lower disease pressure communities may diverge more from each other. These findings do not support the convergence scenario, and instead, support the spatial variation and stochastic variation scenarios, in which community variation is impeding communities from converging towards a similar community state under low disease pressure. Overall, our three-year experiment suggests that foliar fungal diseases not only can maintain or change plant diversity, but also can shape the spatiotemporal dynamics of plant community composition.

Our disease-reduction treatments did not lead to differences in plant community composition. Thus, although plant communities responded to the fungicide treatments in terms of diversity, as shown by richness and Hill-diversity metrics, plant species did not tend to respond in a consistent way that would result in convergence of plant communities within a treatment group (i.e., not consistent with the convergence scenario). The lack of convergence within treatment groups could be related to differences in disease pressure between species and among individuals [41]. For example, generalist parasites could generate fitness differences between multiple host species that have patchy spatial distributions among plots. At the same time, periodic parasite removal could destabilize this system and lead to an increase variation in parasite burden among hosts [42]. It is also possible that plant diversity patterns are related to pre-existing variation in plant communities [43]. While we do not have data on pre-treatment communities of our plots and cannot account for this, we expect the randomization of treatments and high replication of our study should limit these effects. In our system, disease may not consistently favor specific plant species, or we could not detect species-level selection given the large heterogeneity in community composition. Ultimately, spatial variation and stochastic processes that structure communities could be impeding plant communities from converging towards a similar community composition under low disease pressure (i.e., the spatial and stochastic variation scenarios).

Although we did not detect treatment effects on the relative abundance of plant species, we did detect impacts of fungicide on the composition of plant biomass in 2018. Notably, woody-dicot biomass was highest in the plots sprayed with fungicide for 9 months, while tall fescue biomass was highest in the year-round sprayed fungicide plots. These data suggest that there are some compositional differences that are related to specific plant groups, especially tall fescue. Nonetheless, our inferences are limited given that biomass was sorted in one year of the experiment.

Variation in plant community composition among experimental plots was an important component of disease impacts on plant diversity. Specifically, plant community composition was more variable among plots that were treated with the fungicide for longer time periods. Being released from disease appears to amplify variation in community structure in our system, and this could be a result of weaker selection heightening the importance of other processes that govern community structure. Variation in plant community composition among plots could be related to dispersal limitation in our system [44, 45] along with a tendency for communities to exhibit ecological drift over time. Thus, the interplay between multiple processes along with disease can shape diversity patterns over time. Herbivore and predator exclusion experiments have reported similar findings of increased variation in community composition with enemy exclusion and may point to a broader role of natural enemies in reducing variation in community composition [20, 46, 47]. Importantly, any impact disease may have on host community composition could have long lasting effects on host community

assembly and set the stage for historical contingency in local communities and the formation of alternative community states.

In our experiment, plant communities exposed to fungicide for nine months a year showed the greatest variation in community structure among plots. This result was unexpected given that changes in plant biomass were comparable to the always sprayed plots and disease was lowest in the always sprayed plots. The consistent effects of the nine month fungicide treatment on communities could suggest that exposure to disease for a short time in the growing season, potentially corresponding to the phenology of other plant species, could shift plant demographics in unexpected ways. In contrast, the always sprayed fungicide plots showed treatment effects that varied over time and community variation was greatest at the end of the experiment. The interactive effects of fungicide treatment over time on plant community variation could be related to potential non-target effects of the fungicide or shifts in other variables we did not measure.

Over the three years of this experiment, plant communities changed considerably, starting as primarily tall fescue dominated and becoming relatively more diverse. The initial shift in plant communities may have resulted from the selective pruning prior to the experiment to maintain fescue dominance and from a shift in the timing of annual mowing: the experiment was mowed the summer before the experiment was implemented (i.e., in 2016), and thereafter, was mowed during the late fall (after the end of the disease epidemics). Mowing later in the year may have allowed species other than tall fescue, particularly dicots, to grow undisturbed through the summer and fall. This could potentially increase plant establishment from seed, survivorship, growth from root stocks, and size at the end of the growing season, all of which could have led to increased plant diversity over time.

Although plant communities change over time, disease reduction consistently lowered plant richness in our experiment. Hill-Shannon and Hill-Simpson diversity metrics, which place weight on species abundance, revealed treatment effects that varied over time. In the first year of the experiment, Hill-Shannon and Hill-Simpson diversity were similar among fungicide treatment groups. This result reflects the high initial dominance of tall fescue among all plots, but may also be related to an underestimation of plant diversity in the first year. In the first year, the plant community survey was performed a month later (November) than in subsequent years (October). By November, some species may have senesced and not been detected. In subsequent years, after tall fescue declined in both absolute and relative abundance across plots, Hill-Shannon and Hill-Simpson diversity did differ among fungicide treatments and diversity was generally lower in fungicide-treated plots. Foliar fungal disease may increase (as shown here), decrease, or have no effect on plant diversity [9, 48, 49]. These effects of disease on host diversity may interact with host temporal dynamics and further alter the trajectory of host community assembly [6, 11, 32, 50, 51]. In our system, the initial decline of the dominant grass, tall fescue, along with the colonization, extinction, and shifts in abundances of other plants that were potentially released from disease may have resulted in the observed effects on plant diversity over time.

Plants are exposed to a diverse array of parasites that impact their fitness [5, 14], so plant diversity may further be affected by changes in parasite composition [10]. Variation in fungicide impacts could occur via additive effects of disease burdens [52] and/or shifts in parasite composition [10]. In our experiment, treating plant communities with fungicide for a longer time generally reduced disease more. However, differences in the overall disease burden associated with the fungicide treatments did not result in further differences in plant richness among treatment groups. Some of these results could be linked to host specificity [41]. Specifically, the relative contribution of generalist and specialist parasites and their respective impacts on host communities may change as parasite communities assemble under different fungicide

exposure rates. Our measures of host specificity in this system are limited, although similar disease symptoms can be observed on multiple host species (but we cannot rule out cryptic specialization), so it is unclear how host specificity contributed to our results.

Despite timing the fungicide treatments to fit the phenology of the foliar fungal parasites in our system, the treatments had little effect on parasite community composition, so our experiment was not able to rigorously test effects of parasite community composition on plant diversity. In addition to altering disease burdens, changing the infection history of the host population throughout the growing season could have long term consequences for parasite community assembly [53, 54] and disease transmission [26]. While we could not test effects of parasite community composition, the reduction of plant diversity under fungicide treatment, with relatively little variation in effects among the three fungicide treatments, suggests that the effects of fungicide treatment on plant diversity were mediated by something shared among fungicide treatments. For example, in all treatments, fungicide reduced disease in spring and early summer, key seasons for growth of tall fescue.

Interactions between plant communities and disease are reciprocal in nature; disease plays a role in mediating host diversity [4, 5, 9] and at the same time disease transmission and diversity are also driven by host community structure [29, 55]. These feedbacks between disease and host communities may occur on different timescales [29]. For example, in our system, disease diversity and abundance tend to peak later in the growing season, while some disease impacts on host communities (e.g., richness effects) appear to occur earlier, in spring and early summer, as indicated by the similar effects of the different fungicide treatments on plant richness. Therefore, differences in disease burden/abundance may not be directly related to disease impacts on host communities. Scenarios like this may occur if disease diversity or abundance peaks later or earlier than critical time points of host growth or colonization, which help shape communities. Moving forward, incorporating more temporal components, such as varying fungicide treatments within a growing season and monitoring community responses at that time scale, may provide further insight into these complex dynamics.

Along with its impacts on host diversity, disease can reduce plant biomass and contribute to variation in ecosystem productivity [9, 48, 56]. Here, exposure to fungicide and consequently lower levels of disease throughout the growing season generally led to greater plant community biomass. In addition, a longer duration of fungicide exposure further increased plant biomass. Overall, these impacts of disease on plant community biomass were strong and resulted in a ~50% decline in productivity in certain years. Such prominent changes in biomass and primary productivity could have further implications for other consumers in our system. The strong response of plant biomass to differences in disease in our field experiment provides another important example of the effects of diseases on ecosystems [48, 56, 57].

Similar to other natural enemies [20, 46], parasites interact with their hosts in ways that may shape the relative importance of deterministic and stochastic processes throughout community assembly. By exploring three scenarios (convergence, spatial variation, and stochastic variation), we show that the effects of disease on host community structure depend on multiple processes. While disease reduction did not result in a particular signature of deterministic community assembly (i.e., plant communities did not converge in composition within a treatment group), disease did reduce overall variation in community structure. When disease was reduced, communities tended to become less diverse and more variable, suggesting that the relative importance of stochastic processes may increase under lower disease (i.e., under weaker ecological selection). Given that disease could be an agent of selection in many systems, resolving its role in either reducing or amplifying variation among communities may yield insights into its consequences for the trajectory of community assembly. Taken together, our study suggests that while disease impacts on host communities may be chiefly driven by

selection in conjunction with other deterministic processes like competition, how disease shapes communities over time also can depend on stochastic processes structuring those communities.

## Supporting information

**S1 Table. Log response ratios (LRR) of fescue disease AUDPS, area under the disease progress stairs, in response to fungicide treatments.** Reported are % change in AUDPS relative to the control, bootstrapped LRR and 95% confidence intervals.
(DOCX)

**S1 Fig. Ranked log transformed abundance (A) and raw-untransformed abundance (B) of all plant species across years in experimental plots.** Relative abundance (% cover) was averaged across all 64 experimental plots for a given year. Tall fescue (*Lolium arundinaceum*) was the dominant plant species each year.
(TIF)

**S2 Fig. Disease of tall fescue, quantified as the cross-species disease burden (AUDPS) based on prevalences of the three focal diseases (anthracnose, brown patch, and crown rust), was strongly reduced by fungicide treatments and these treatment effects varied across years.** Plotted are observed treatment means and their 95% confidence intervals, and smaller points represent the raw data that are jittered to show the distribution of the data. Lower case letters denote groups based on Tukey post-hoc tests using all pairwise comparisons to compare treatment effects over time.
(TIF)

**S3 Fig. The effects of fungicide treatments on disease burden (AUDPS) of three diseases of tall fescue (anthracnose, brown patch, crown rust).** Large points indicate treatment means with their 95% confidence intervals, while smaller points are the raw data. Raw data points are jittered to show the distribution of the data.
(TIF)

**S4 Fig. The relative contribution of anthracnose, brown patch, and crown rust to the overall cross-disease disease burden (AUDPS).** The disease anthracnose was the chief contributor to total disease across all treatments and years.
(TIF)

**S5 Fig. In 2018, the aboveground biomass of certain plant groups differed among fungicide treatments (MANOVA, treatment: $F_{3,60} = 4.23$, Wilks $\wedge = 0.20$, $p < 0.0001$).** Specifically, differences among treatments in 2018 plant biomass were attributed to tall fescue (ANOVA, treatment: $F_{3,60} = 11.47$, $p < 0.0001$, $\eta^2 = 0.36$) and woody dicots ($F_{3,60} = 3.32$, $p = 0.03$, $\eta^2 = 0.14$). Litter ($F_{3,60} = 1.32$, $p = 0.28$), non-fescue monocots ($F_{3,60} = 1.87$, $p = 0.14$), and non-woody dicots ($F_{3,60} = 1.72$, $p = 0.17$) were not affected by fungicide treatments in 2018. Letters denote post hoc comparisons between treatments within each plant biomass group. Plotted are observed treatment means and their associated 95% CI, and smaller points represent the raw data that are jittered to show the distribution of the data.
(TIF)

**S6 Fig. Plant community dissimilarity patterns varied across years, and were not affected by fungicide treatment.** Shown are results of nonmetric multidimensional scaling of plant communities based on Bray-Curtis distances. All data are plotted on the same axes for A, C, E to show all community dissimilarity patterns. Panels B, D, F, are zoomed in to show the core

community patterns. Species vectors show the distribution of plant taxa among communities. NMDS Stress = 0.12.
(TIF)

**S7 Fig. Plant community composition varied across years and was not impacted by fungicide treatments.** Plotted are treatment centroids of nonmetric multidimensional scaling based on Bray-Curtis distances and their 95% confidence intervals. NMDS Stress = 0.12.
(TIF)

## Acknowledgments

For their assistance with field work, we thank: Anita Simha, Brandon Wheeler, Matthew Carey, Julie Long, Storm Crews, Charles Muirhead, Jordan Link, Claire Thefaine, Safiyyah Motaib, Julia Knorr, and Ryan Cook. We thank Peter Morin and members of the Morin lab for their feedback. The findings and conclusions of this publication are those of the authors and should not be construed to represent any official USDA or U.S. Government determination or policy.

## Author Contributions

**Conceptualization:** Fletcher W. Halliday, Charles E. Mitchell.

**Data curation:** Brooklynn N. Joyner.

**Formal analysis:** Rita L. Grunberg.

**Funding acquisition:** Charles E. Mitchell.

**Investigation:** Fletcher W. Halliday, Robert W. Heckman, Brooklynn N. Joyner, Kayleigh R. O'Keeffe.

**Methodology:** Fletcher W. Halliday, Robert W. Heckman, Brooklynn N. Joyner, Kayleigh R. O'Keeffe.

**Project administration:** Fletcher W. Halliday, Brooklynn N. Joyner, Kayleigh R. O'Keeffe, Charles E. Mitchell.

**Resources:** Charles E. Mitchell.

**Supervision:** Charles E. Mitchell.

**Visualization:** Rita L. Grunberg.

**Writing – original draft:** Rita L. Grunberg.

**Writing – review & editing:** Rita L. Grunberg, Fletcher W. Halliday, Robert W. Heckman, Brooklynn N. Joyner, Kayleigh R. O'Keeffe, Charles E. Mitchell.

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
