## [Decision Letter · Decision Letter 0]

27 Jun 2023

PONE-D-23-04650Disease decreases variation in host community structure in an old-field grasslandPLOS ONE 

Dear Dr. Grunberg,

Thank you for submitting your manuscript to PLOS ONE. After careful consideration, we feel that it has merit but does not fully meet PLOS ONE’s publication criteria as it currently stands. Therefore, we invite you to submit a revised version of the manuscript that addresses the points raised during the review process.

The reviewers referred to several points that need further clarification or improvement. On the one hand, these points relate to the introduction, where the issue of convergence versus divergence of plant communities could be better elaborated and related to starting conditions and to the role of dominant plant species. Likewise, the concept of ecological selection should be introduced to readers not familiar with it, and related to host specifity of pathogens. Finally, the hypotheses seem to be a bit vague, and should be used to deduce expectations related to the specific study system.

On the other hand, there are a couple of points regarding the interpretation results and their discussion: To what extent may the observer error regarding cover estimates have influenced the results? Can the initial weeding effort have caused the change in variation among plant communities? Which non-target effects of fungicide can be expected and may they have caused the weaker effects of the year-round treatment? In addition, reviewers suggested that treatment effects on tall fescue biomass can be used to quantify responses in plant community composition, and that fungicide effects on composition of the pathogen community should be statistically evaluated.

We look forward to receiving your revised manuscript.

Kind regards,

Harald Auge

Academic Editor

PLOS ONE

Journal Requirements:

2. In your Methods section, please provide additional information regarding the permits you obtained for the work. Please ensure you have included the full name of the authority that approved the field site access and, if no permits were required, a brief statement explaining why

"This work was supported by the NSF-USDA joint program in Ecology and Evolution of Infectious Diseases (USDA-NIFA AFRI grant 2016-67013-25762). "

"This work was supported by the NSF-USDA joint program in Ecology and Evolution of Infectious Diseases (USDA-NIFA AFRI grant 2016-67013-25762) awarded to CEM. "

"This work was supported by the NSF-USDA joint program in Ecology and Evolution of Infectious Diseases (USDA-NIFA AFRI grant 2016-67013-25762) awarded to CEM. "           

Reviewers' comments:

Reviewer's Responses to Questions

**Comments to the Author**

1. Is the manuscript technically sound, and do the data support the conclusions?

Reviewer #1: Yes

Reviewer #2: Yes

2. Has the statistical analysis been performed appropriately and rigorously? 

Reviewer #1: Yes

Reviewer #2: Yes

3. Have the authors made all data underlying the findings in their manuscript fully available?

Reviewer #1: Yes

Reviewer #2: Yes

4. Is the manuscript presented in an intelligible fashion and written in standard English?

Reviewer #1: Yes

Reviewer #2: Yes

5. Review Comments to the Author

Reviewer #1: In the manuscript “Disease decreases variation in host community structure in an old-field grassland”, Grunberg et al. describe a three-year field experiment in which foliar fungal pathogens are suppressed with fungicide treatments and impacts on the plant community are measured. They hypothesized that suppressing disease would reduce the importance of ecological selection and elevate the importance of stochastic processes, leading to shifts in plant community diversity and variation. They found that suppressing disease decreased plant richness and diversity, increased plant biomass, and increased variation in community composition among plots. I think the broad community ecology themes, rigorously conducted experiment, and clear writing will make the manuscript an excellent contribution to PLOS ONE once the hypotheses are more clearly described and connected to the study system (see major comments).

Major comments

1. The introduction was generally difficult for me to follow. One suggestion is to include more specific examples of biological systems because these can help make general statements more tangible. Here are some specific places where I got stuck:

a. I’m confused about the convergence hypothesis. I understand the sentence L52-55, but then I get lost in the following sentences. Questions that come to mind when reading are: What are the starting set of communities you expect to converge? Is disease expected to drive convergence or divergence? How does convergence relate to suppression of a dominant species?

b. L61-72: What’s the relationship between disease as an agent of ecological selection and host-specificity of the parasite(s)? Describing this would help calrify the hypothesis. It would also clarify what you mean by ecological selection for readers who are less familiar with Vellend’s framework. The question also comes up for me throughout the paper due to the dominance of tall fescue and the generality of the parasites – I’m not quite sure how to apply the hypothesis to this system.

c. L90-97: This is a great opportunity to bring in specifics of the system and map them on to your hypotheses. As they’re stated here, the hypotheses are too vague for me to form expectations for the results. For example, maybe you expect parasites contribute to either plant diversity or the dominance of tall fescue (the role of ecological selection described in the abstract)? Maybe one of the diseases is expected to have a stronger impact on tall fescue than the others?

2. L200-209: If one of the goals of the experimental design was to manipulate the composition of parasites (L86-88), it seems like there should be a statistical test to evaluate whether the fungicide treatments affected the composition. In Figures S3 and S4, it looks like the three diseases were differentially affected by the fungicide treatments, even if anthracnose remained dominant (L268-274).

3. It’s unclear to me why the result about the treatment effect on tall fescue biomass in 2018 (Fig. S5) isn’t used as a measure of parasites affecting plant community composition. It seems like a very informative result.

4. Researchers can vary in plant ID skills and percent cover estimation, which influence the community composition and variation results. If one person did not collect all plant community data, a description of how observer error may influence the results or was accounted for would be helpful.

Minor comments

1. L47+: Are these statements meant to describe only plant communities, or do they include non-plant communities as well?

2. L120-124: A more explicit description of which diseases you expect to be suppressed with each treatment would be helpful. If possible, a brief description of relevant aspects of the parasite life histories (in the methods or discussion) would provide context for the experimental treatments and perhaps an explanation for deviations from the expected effects of treatments on parasite composition (e.g., alternative hosts or life stages not represented in Fig. 1).

3. L156-157: What creates the 2-m buffer that surrounds the entire experimental area?

4. L197-199: Is the cross-disease disease burden calculated from data collected as the proportion of leaves with any of the diseases? Or were the disease prevalences of the three diseases somehow combined after data collection?

5. L241-243: The impacts of plant pathogens can be quite subtle, so it is probably more realistic to expect the plant community to shift due to growth and reproduction than colonization and loss of species. Because you measured relative cover rather than only presence/absence, you were able to capture all these processes. I recommend re-wording the hypothesis to reflect this.

6. L259: Should be Fig. S2 instead of S3?

7. L279+: Indicate that you’re referring to all of the fungicide treatments.

8. L348-351: Potential discussion point: Do you think the decline in tall fescue is a result of clipping co-dominant species in 2017, but not the following years?

9. L361-364: From Fig. 5, it looks like this only applies to 2019, but maybe the statistical test was performed on all years?

10. If you’re able to (based on field observations), it would be helpful to mention in the discussion whether percentage of leaf area with symptoms seemed to be correlated with percentage of leaves infected, as this alternative measure of disease burden is used in other studies and may yield different results for parasite composition.

11. It’s worth noting in the discussion that there’s no pre-treatment measure of richness or diversity, so it’s unclear how much the reduction in richness is due to the treatment vs. existing plot differences. The one-year lag in treatment effect of the diversity metrics help address this concern, but it’s still an unknown.

Reviewer #2: Comments to Author:

The manuscript by Grunberg et al. “Disease decreases variation in host community structure in an old-field grassland” presents multi-year data from an experiment assessing the role of fungal disease on plant community structure, diversity, and biomass. Reducing disease caused plant communities to decrease in diversity and increase in biomass. Additionally, they found that fungicide applications increased the variation in plant community structure, highlighting a potential important role of plant disease in community assembly of natural plant communities.

These findings have important consequences for understanding plant-microbe interactions and how deterministic processes such as disease control plant community dynamics. The framing, methods, results, and conclusions are all clearly written and tell an interesting story. I do feel that some aspects of the methods and results could be elaborated on in the discussion.

General comments:

1. As part of the methods, the plant communities were manipulated to increase the dominance of tall fescue (by clipping shoots of other abundant species). This suggests that the plant communities across all experimental plots at the beginning of the experiment are starting at a more homogenous state than in later years of this experiment. The authors argue that the change in mowing regime may be the reason for increased variation through time, but the initial weeding of certain plant species only in the first year could just as likely be a causative agent of this change in variation over time. I do not think this point relates much to the results/conclusions of fungicide impacts on variation in community structure, but it is important to acknowledge the potential impact of this methodological approach.

2. The authors found that the 9 months fungicide treatment had consistent (and higher) impacts on variation in community structure compared to the year-round fungicide treatment (which had no effects in first year but later had impacts on distance to centroid). Given that the year-round fungicide reduced disease the most and increased biomass to the same degree as the 9 months treatment, I would have liked to see some discussion on why the year-round treatment was weaker than the 9months treatment when looking at variation in community structure. Does this suggest that there could be non-target effects of the fungicide when applied year-round? Despite reducing disease and increasing biomass, the year-round fungicide may be altering some other aspect of the microbiome during the non-growing season or may be altering soil chemistry in such a way that caused communities to be more homogenous compared to 9 month treatment?? These thoughts cannot be tested with the existing data, but I think a mention of this divergence among the two most extreme fungicide treatments for the most novel measurement in the study is worth addressing in the discussion.

Minor comments.

1. L137-138: You mention that the fungicide is applied at the recommended rate, but then do not give the details on the volume or frequency, so it is unclear what you mean by rate. Do you just mean the 2 week intervals of application? If you mean volume, then add details or remove this mention of recommended rate.

2. Figure 1 Legend: AUDPS is not defined in the legend. The reader would have to go into the main text to understand this acronym so please define in the legend as well.

3. L156-158: Is the 16x4 array of experimental plots similar to the idea of blocks? If so is it worth including a Block structure in your models as random effects? The landscape did not look extremely heterogenous across arrays (from Figure 2), but this mention of the arrays made me think about block structures and whether they were addressed in analyses (maybe not necessary?).

4. L166-167: Litter and bare ground were estimated in addition to plant species, but this data is not presented in the plant abundance supplemental tables. I was also curious then if litter and bare ground were included in plant community composition analyses? Were litter and bare ground excluded from plant community analyses?

5. L342: This first sentence is a bit vague. The fungicide applications had no effect on plant composition – so then the communities are essentially the same in composition within a given sampling year. So the point of communities not converging is based on the differences in composition among years? In terms of fungicide application, plant communities converged (or never diverged) across the control and fungicide plots.

6. L356-358: Same as my previous comment, if the fungicide plots do not differ in composition compared to controls, then does that not suggest that composition is uniform across experimental treatments?

7. L367-371: This interaction is what I referred to in my general comments. Any ideas on why the 9 month treatment is more consistent (and stronger) than year-round fungicide for impacts on variation in community composition?

8. L402-403: High variation in community composition could be due to extinctions/colonization, but could also be due to shifting abundances (still potentially due to stochastic processes).

9. L406-408: If disease does not favor specific plant species, are you suggesting that generalist pathogens are the major players here? Is it worth mentioning spill-over effects and the contribution of host-specific versus generalist pathogens given what you find in terms of plant biomass responses versus plant composition?

10. L415-416: “Herbivore and predator exclusion experiments have reported similar findings to our study” do you mean the finding of increased variation in composition with enemy exclusion? Please be more specific to which finding you refer to.

11. L423-426: These lines are related to my other general comment. Why would the selective pruning of certain species in the first year not also lead to increased variation as time continues and the pruning stops?

12. L476-477: You have great phenology data on the disease in this manuscript, so it makes sense to me that the next step would be including time-series data on the plant communities (within-growing season dynamics of the plants) to match that resolution in data to better address this gap in knowledge. More a thought than something you need to include in this discussion.

13. Figure S6: There are not labels for the panels (supposed to be labels for panels A-F). Please include to match the legend.

6. PLOS authors have the option to publish the peer review history of their article (what does this mean?). If published, this will include your full peer review and any attached files.

Reviewer #1: No

Reviewer #2: No

---

## [Author Response · Author response to Decision Letter 0]

4 Aug 2023

Dear Editor,

Thank you for the opportunity to submit a revised version of our manuscript titled “Disease decreases variation in host community structure in an old-field grassland.” We thank the reviewers for their helpful comments and suggestions and have revised the manuscript in response to their feedback. 

We have provided a response letter to the comments from each reviewer with line references below in bold italics and a tracked changes version of our manuscript. 

In addition, below is a list of the requested formatting changes that were completed.

1.In the methods section, we now provide our field permit: Duke Forest Teaching and Research Laboratory Research Registration number: R1617-434 (lines 113).

2.As requested, we removed the funder information from the Acknowledgements section. 

3.Our current Funding Statement should be listed as: "This work was supported by the NSF-USDA joint program in Ecology and Evolution of Infectious Diseases (USDA-NIFA AFRI grant 2016-67013-25762) awarded to CEM and an Ambizione grant (PZ00P3_202027) from the Swiss National Science Foundation awarded to FWH.” 

4.Amended Role of Funder statement to be used in the manuscript: "The funders had no role in study design, data collection and analysis, decision to publish, or preparation of the manuscript."

5.Our data and code availability statement now includes a DOI and is available through zenodo at 10.5281/zenodo.6980274

6. We moved figure captions for the Supporting Information files to the end of the manuscript. 

We feel these changes have improved the quality of our manuscript, and we hope that you will now find this work suitable for PLOS ONE. Thank you for your efforts in the publication of our research.

Sincerely, 

Rita Grunberg

Review Comments to the Author

Reviewer #1: In the manuscript “Disease decreases variation in host community structure in an old-field grassland”, Grunberg et al. describe a three-year field experiment in which foliar fungal pathogens are suppressed with fungicide treatments and impacts on the plant community are measured. They hypothesized that suppressing disease would reduce the importance of ecological selection and elevate the importance of stochastic processes, leading to shifts in plant community diversity and variation. They found that suppressing disease decreased plant richness and diversity, increased plant biomass, and increased variation in community composition among plots. I think the broad community ecology themes, rigorously conducted experiment, and clear writing will make the manuscript an excellent contribution to PLOS ONE once the hypotheses are more clearly described and connected to the study system (see major comments).

Major comments

1. The introduction was generally difficult for me to follow. One suggestion is to include more specific examples of biological systems because these can help make general statements more tangible. Here are some specific places where I got stuck:

a. I’m confused about the convergence hypothesis. I understand the sentence L52-55, but then I get lost in the following sentences. Questions that come to mind when reading are: What are the starting set of communities you expect to converge? Is disease expected to drive convergence or divergence? How does convergence relate to suppression of a dominant species?

Author: Thank you for the feedback. We unpacked these ideas further and made the connections more specific to help explain our ideas on convergence within treatments. As a point of clarification, we now restate that we are considering within-treatment convergence. See lines 54-66. Text is also pasted below. 

‘This could lead to community convergence because disease is consistently reducing the abundance of the better competitor, which may lead to an increase in abundance of other competing species generating communities with a similar composition. Importantly, the strength of convergence may also depend on the distribution and abundance of species within communities. For example, initial variation in species composition among communities can lead to variation in the degree of selection by natural enemies, impeding host communities from converging towards a similar composition [15]. Specifically, if a host species that is impacted heavily by disease is patchy in their distribution, then the strength of selection by that disease may vary among communities. Further, when stochastic processes are key drivers of community assembly, the colonization, extinction, and abundance of species could vary greatly among communities and drive differences in the impacts of disease over space and tme.’

b. L61-72: What’s the relationship between disease as an agent of ecological selection and host-specificity of the parasite(s)? Describing this would help clarify the hypothesis. It would also clarify what you mean by ecological selection for readers who are less familiar with Vellend’s framework. The question also comes up for me throughout the paper due to the dominance of tall fescue and the generality of the parasites – I’m not quite sure how to apply the hypothesis to this system.

Author: This is a very important point. First, we added a definition of selection, i.e., biotic and abiotic factors that contribute to species filtering, in the first sentence of the introduction (line 46). We further explain the role of host specificity (lines 83-86) and its relationship with disease as an agent of selection. This point is also reflected throughout the discussion (lines 438-441; 512-517). We also clarified that our ability to describe the level of host specificity in this study is limited, because we cannot exclude more cryptic specialization (lines 133-135). 

c. L90-97: This is a great opportunity to bring in specifics of the system and map them on to your hypotheses. As they’re stated here, the hypotheses are too vague for me to form expectations for the results. For example, maybe you expect parasites contribute to either plant diversity or the dominance of tall fescue (the role of ecological selection described in the abstract)? Maybe one of the diseases is expected to have a stronger impact on tall fescue than the others?

Author: At the end of the introduction, we added text that is more specific to our system to better set up expectations. A key limitation is that our previous understanding of disease impacts was largely based at the host individual level (potted plants). The exception was tall fescue invading the plots from previous experiments, but fescue was just one of many players there, and also that population context is very different from the long-established population in this experiment. 

Pasted below in the text added to the introduction. Lines 103-113. 

‘Our experiment focused on disease impacts within a long-established grassland community dominated by tall fescue. Given the high dominance of tall fescue, disease impacts on this host species may contribute to differences in plant diversity in our system. At one end, low disease pressure could promote convergence among the low disease communities when plant species consistently change in their abundance. For example, if tall fescue is relatively less impacted by disease, then disease reduction may favor the establishment of other plant species that are generally more impacted. This could result in distinct low and high disease plant communities. In this scenario, communities may also fail to generate distinct compositions associated with their exposure to disease because stochastic processes could generate high variation in community structure among experimental communities.’

2. L200-209: If one of the goals of the experimental design was to manipulate the composition of parasites (L86-88), it seems like there should be a statistical test to evaluate whether the fungicide treatments affected the composition. In Figures S3 and S4, it looks like the three diseases were differentially affected by the fungicide treatments, even if anthracnose remained dominant (L268-274).

Author: As suggested, we modeled each disease’s relative contribution to the cross-disease AUDPS to assess changes in parasite composition. Our model indicates there was a change (although small) in parasite composition, and this was due to changes in the crown rust (Treat, p = 0.002) and this effect was consistent across years (treat * time p =0.9) 

3. It’s unclear to me why the result about the treatment effect on tall fescue biomass in 2018 (Fig. S5) isn’t used as a measure of parasites affecting plant community composition. It seems like a very informative result.

Author: We agree this is an informative result and now highlight it as another measure of disease impacts on host community composition in the discussion. See lines 448-454. 

4. Researchers can vary in plant ID skills and percent cover estimation, which influence the community composition and variation results. If one person did not collect all plant community data, a description of how observer error may influence the results or was accounted for would be helpful.

Author: All plant IDs and cover estimations were done by a single individual throughout the experiment. We now state this in the methods section. See lines 192. 

Minor comments

1. L47+: Are these statements meant to describe only plant communities, or do they include non-plant communities as well?

Author: All these statements apply to both animal and plant communities. 

2. L120-124: A more explicit description of which diseases you expect to be suppressed with each treatment would be helpful. If possible, a brief description of relevant aspects of the parasite life histories (in the methods or discussion) would provide context for the experimental treatments and perhaps an explanation for deviations from the expected effects of treatments on parasite composition (e.g., alternative hosts or life stages not represented in Fig. 1).

Author: A more detailed description of the biology of the parasites and the aimed impacts of the fungicide on specific diseases is now included in the methods. Text also pasted below: 

Lines 124-132: ‘These diseases are largely generalists and observed on other common plant species in this system [24], although the same disease symptom may be caused on different plant species by different cryptic species of fungi [25]. Anthracnose and brown patch are hemibiotrophic and necrotrophic pathogens, respectively, and have been observed to infect a variety of host species in this system. However, crown rust is an obligate biotroph that tends to be more host specific but has been observed to infect some additional hosts in this system.’

Lines 135-140: ‘Our fungicide treatment regimens correspond with the peak of the seasonal epidemics of these diseases, so that different fungicide treatments would shift the composition of disease. Treatments varied in their duration of fungicide exposure including: (1) no fungicide (control) to represent ambient levels of all diseases, (2) fungicide until mid-July (approximately 7 months, soon after the typical start of brown patch epidemics) to reduce the seasonal window of anthracnose infections, (3) fungicide until mid-September (approximately nine months, soon after the typical start of crown rust epidemics) to reduce both the anthracnose and brown patch seasonal infection window, and (4) year-round application of fungicide to reduce all diseases.’

3. L156-157: What creates the 2-m buffer that surrounds the entire experimental area?

Author: Here, we clarified that the buffer is a 2-m mowed section around the experimental area 

4. L197-199: Is the cross-disease disease burden calculated from data collected as the proportion of leaves with any of the diseases? Or were the disease prevalences of the three diseases somehow combined after data collection?

Author: Thank you for pointing this out. We clarified that we used the proportion of leaves infected by any disease to calculate cross-disease AUDPS. 

5. L241-243: The impacts of plant pathogens can be quite subtle, so it is probably more realistic to expect the plant community to shift due to growth and reproduction than colonization and loss of species. Because you measured relative cover rather than only presence/absence, you were able to capture all these processes. I recommend re-wording the hypothesis to reflect this.

Author: This is a good point. We clarified that we are capturing multiple processes that will shape these communities. 

6. L259: Should be Fig. S2 instead of S3?

Author: Correct. Fixed (now line 294). 

7. L279+: Indicate that you’re referring to all of the fungicide treatments.

Author: Fixed 

8. L348-351: Potential discussion point: Do you think the decline in tall fescue is a result of clipping co-dominant species in 2017, but not the following years?

Author: The clipping could contribute to the decline of tall fescue in the following years, and this point is now acknowledged in the discussion (lines 480-482). Generally, the amount of clipping done was minimal, so we do not expect it to contribute substantially to our results. 

9. L361-364: From Fig. 5, it looks like this only applies to 2019, but maybe the statistical test was performed on all years?

Author: To clarify, the statistical test was performed on all years. 

10. If you’re able to (based on field observations), it would be helpful to mention in the discussion whether percentage of leaf area with symptoms seemed to be correlated with percentage of leaves infected, as this alternative measure of disease burden is used in other studies and may yield different results for parasite composition.

Author: We agree this would be valuable, but we do not have data to address this point. 

11. It’s worth noting in the discussion that there’s no pre-treatment measure of richness or diversity, so it’s unclear how much the reduction in richness is due to the treatment vs. existing plot differences. The one-year lag in treatment effect of the diversity metrics help address this concern, but it’s still an unknown.

Author: This point is mentioned in the discussion. See lines: 433-437. We also suggest that given the randomization of treatments and high replication this should prevent one treatment from, by chance, being assigned to all the plots that have a certain composition.

Reviewer #2: Comments to Author:

The manuscript by Grunberg et al. “Disease decreases variation in host community structure in an old-field grassland” presents multi-year data from an experiment assessing the role of fungal disease on plant community structure, diversity, and biomass. Reducing disease caused plant communities to decrease in diversity and increase in biomass. Additionally, they found that fungicide applications increased the variation in plant community structure, highlighting a potential important role of plant disease in community assembly of natural plant communities.

These findings have important consequences for understanding plant-microbe interactions and how deterministic processes such as disease control plant community dynamics. The framing, methods, results, and conclusions are all clearly written and tell an interesting story. I do feel that some aspects of the methods and results could be elaborated on in the discussion.

General comments:

1. As part of the methods, the plant communities were manipulated to increase the dominance of tall fescue (by clipping shoots of other abundant species). This suggests that the plant communities across all experimental plots at the beginning of the experiment are starting at a more homogenous state than in later years of this experiment. The authors argue that the change in mowing regime may be the reason for increased variation through time, but the initial weeding of certain plant species only in the first year could just as likely be a causative agent of this change in variation over time. I do not think this point relates much to the results/conclusions of fungicide impacts on variation in community structure, but it is important to acknowledge the potential impact of this methodological approach.

Author: We agree this is an important methodological topic to acknowledge. We do highlight the potential effects of clipping on the decline in tall fescue (and increase variation in community composition). Generally, the amount of clipping done was minimal, so we do not expect it to contribute substantially to our results. 

Text pasted below (lines 479-482): 

‘Over the three years of this experiment, plant communities changed considerably, starting as primarily tall fescue dominated (as per our design) and becoming relatively more diverse. These shifts in plant communities may have resulted from the selective pruning prior to the experiment to maintain fescue dominance and in part from a shift in the timing of annual mowing: the experiment was mowed the summer before the experiment was implemented (i.e., in 2016), and thereafter, was mowed during the late fall (after the end of the disease epidemics).’

2. The authors found that the 9 months fungicide treatment had consistent (and higher) impacts on variation in community structure compared to the year-round fungicide treatment (which had no effects in first year but later had impacts on distance to centroid). Given that the year-round fungicide reduced disease the most and increased biomass to the same degree as the 9 months treatment, I would have liked to see some discussion on why the year-round treatment was weaker than the 9months treatment when looking at variation in community structure. Does this suggest that there could be non-target effects of the fungicide when applied year-round? Despite reducing disease and increasing biomass, the year-round fungicide may be altering some other aspect of the microbiome during the non-growing season or may be altering soil chemistry in such a way that caused communities to be more homogenous compared to 9 month treatment?? These thoughts cannot be tested with the existing data, but I think a mention of this divergence among the two most extreme fungicide treatments for the most novel measurement in the study is worth addressing in the discussion.

Author: We added a paragraph to the discussion to address this important point. A priori, we did not expect this result. We think this could be related to the phenology of the hosts. It could be that exposure to disease at the end of the growing season is having some unexpected effect on plant demographics, or a bit of disease has more impact than having almost no disease in the system. 

Text pasted below (from lines 468-478):

‘In our experiment, plant communities exposed to fungicide for nine months a year showed the greatest variation in community structure among plots. This result was unexpected given that changes in plant biomass were comparable to the always sprayed plots and disease was lowest in the always sprayed plots. The consistent effects of the nine month fungicide treatment on communities could suggest that exposure to disease for a short time in the growing season, potentially corresponding to the phenology of other plant species, could shift plant demographics in unexpected ways. In contrast, the always sprayed fungicide plots showed treatment effects that varied over time and community variation was greatest at the end of the experiment. The interactive effects of fungicide treatment over time on plant community variation could be related to potential non-target effects of the fungicide or shifts in other variables we did not measure.’

Minor comments.

1. L137-138: You mention that the fungicide is applied at the recommended rate, but then do not give the details on the volume or frequency, so it is unclear what you mean by rate. Do you just mean the 2 week intervals of application? If you mean volume, then add details or remove this mention of recommended rate.

Author: Thank you for pointing this out. This refers to the recommended rate provided by the fungicide manufacturer. The manuscript now specifies the concentration applied (20 g/1 gal of water); the volume applied was adjusted as vegetation grew and senesced to cover all leaves with the fungicide. We also clarified this in the main text (lines 161-163). 

2. Figure 1 Legend: AUDPS is not defined in the legend. The reader would have to go into the main text to understand this acronym so please define in the legend as well.

Author: Fixed. 

3. L156-158: Is the 16x4 array of experimental plots similar to the idea of blocks? If so is it worth including a Block structure in your models as random effects? The landscape did not look extremely heterogenous across arrays (from Figure 2), but this mention of the arrays made me think about block structures and whether they were addressed in analyses (maybe not necessary?). (

Author: There was no spatial blocking with respect to treatments. We fully randomized the treatments within this spatial array. 

4. L166-167: Litter and bare ground were estimated in addition to plant species, but this data is not presented in the plant abundance supplemental tables. I was also curious then if litter and bare ground were included in plant community composition analyses? Were litter and bare ground excluded from plant community analyses?

Author: Litter and bare ground were excluded from the plant community analysis. We also clarified this point in the main text (line 197).

5. L342: This first sentence is a bit vague. The fungicide applications had no effect on plant composition – so then the communities are essentially the same in composition within a given sampling year. So the point of communities not converging is based on the differences in composition among years? In terms of fungicide application, plant communities converged (or never diverged) across the control and fungicide plots.

Author: Thank you for bringing this to our attention. The point of convergence would be that over time, a specific treatment yields a specific community composition. For example, plots that experienced no fungicide would yield a community that contains certain species and plots that experience fungicide year-round would yield a community with another composition. To clarify, the communities weren’t the same, it is just that within a treatment, the communities are so variable so we could not detect statistical differences in relative abundances of plants. In response to this, we frequently changed our language to ‘within-treatment convergence’

6. L356-358: Same as my previous comment, if the fungicide plots do not differ in composition compared to controls, then does that not suggest that composition is uniform across experimental treatments?

Author: The multivariate glm is asking whether the average composition is different from each other. Here, the compositions are so variable from each other (as shown by the distance from the centroid test), that we could not detect a statistical difference between treatments. This is similar to thinking about a levene test and ANOVA. If the within group variance is so high, then between group differences are less likely to be detected. 

7. L367-371: This interaction is what I referred to in my general comments. Any ideas on why the 9 month treatment is more consistent (and stronger) than year-round fungicide for impacts on variation in community composition?

Author: We addressed this important point in the discussion. A priori, we did not expect this result. We think this could be related to the phenology of the hosts. It could be that exposure to disease at the end of the growing season is having some unexpected effect on plant demographics, or a bit of disease has more impact than having almost no disease in the system. See lines 468-478

8. L402-403: High variation in community composition could be due to extinctions/colonization, but could also be due to shifting abundances (still potentially due to stochastic processes).

Author: Good point. We included that changes in abundances could also be driving variation in composition. 

9. L406-408: If disease does not favor specific plant species, are you suggesting that generalist pathogens are the major players here? Is it worth mentioning spill-over effects and the contribution of host-specific versus generalist pathogens given what you find in terms of plant biomass responses versus plant composition?

Author: These are good hypotheses that could apply in some systems, but no, we are not trying to suggest them here. Disease as an agent of selection isn’t favoring a specific plant species, in part because communities are so variable in community structure. It may be that anthracnose and brown patch have a larger host range in this system, so this could be related to spillover to other hosts that we did not survey, but it also may be that there are cryptic species of the fungal pathogens that are specialized on different host species. 

10. L415-416: “Herbivore and predator exclusion experiments have reported similar findings to our study” do you mean the finding of increased variation in composition with enemy exclusion? Please be more specific to which finding you refer to.

Author: We clarified this in the main text. We are referring to the increase variation in enemy exclusion experiments. 

11. L423-426: These lines are related to my other general comment. Why would the selective pruning of certain species in the first year not also lead to increased variation as time continues and the pruning stops?

Author: We address this point as per the main comments.

12. L476-477: You have great phenology data on the disease in this manuscript, so it makes sense to me that the next step would be including time-series data on the plant communities (within-growing season dynamics of the plants) to match that resolution in data to better address this gap in knowledge. More a thought than something you need to include in this discussion.

Author: We agree that this would be an important next step. 

13. Figure S6: There are not labels for the panels (supposed to be labels for panels A-F). Please include to match the legend.

Author: Thank you for pointing this out. We fixed the labels for Figure S6.

---

## [Decision Letter · Decision Letter 1]

31 Aug 2023

PONE-D-23-04650R1Disease decreases variation in host community structure in an old-field grasslandPLOS ONE

Dear Dr. Grunberg,

Thank you for submitting your manuscript to PLOS ONE. After careful consideration, we feel that it has merit but does not fully meet PLOS ONE’s publication criteria as it currently stands. Therefore, we invite you to submit a revised version of the manuscript that addresses the points raised during the review process. First of all, I would like to acknowledge how you addressed the reviewers' comments in the revised manuscript. However, I fully agree with reviewer #1 that there is still one issue that needs further consideration. This refers to how the hypotheses, the underlying assumptions and the system-specific expectations are presented in the introduction. Please restructure and complete these parts of the introduction carefully according to the reviewer's comments. I'm looking forward to reading a revised version soon.

We look forward to receiving your revised manuscript.

Kind regards,

Harald Auge

Academic Editor

PLOS ONE

Journal Requirements:

Reviewers' comments:

Reviewer's Responses to Questions

**Comments to the Author**

1. If the authors have adequately addressed your comments raised in a previous round of review and you feel that this manuscript is now acceptable for publication, you may indicate that here to bypass the “Comments to the Author” section, enter your conflict of interest statement in the “Confidential to Editor” section, and submit your "Accept" recommendation.

Reviewer #1: (No Response)

Reviewer #2: All comments have been addressed

2. Is the manuscript technically sound, and do the data support the conclusions?

Reviewer #1: Yes

Reviewer #2: Yes

3. Has the statistical analysis been performed appropriately and rigorously? 

Reviewer #1: Yes

Reviewer #2: Yes

4. Have the authors made all data underlying the findings in their manuscript fully available?

Reviewer #1: Yes

Reviewer #2: Yes

5. Is the manuscript presented in an intelligible fashion and written in standard English?

Reviewer #1: Yes

Reviewer #2: Yes

6. Review Comments to the Author

Reviewer #1: The authors successfully addressed almost all of my comments and I appreciated their responses. However, I still had to read the introduction a few times to understand the motivating theory/hypotheses. I know I already gave a lengthy comment on this in my last review, so I will try to explain it differently this time. I think that all the ideas are there but are difficult to understand due to unbalanced presentation of hypotheses and implicit assumptions. It also might help to reconsider the order of ideas in the introduction. The two potential outcomes of disease on convergence are presented in an unbalanced way because one is the main conceptual hypothesis and the other is system-specific hypothesis. The “main conceptual hypothesis” is presented on L23-24, L56-58, and L73-75: disease drives convergence of communities because species that are more tolerant/resistant become more dominant. This hypothesis needs more citations and/or explanation of implicit assumptions because it is unclear why the more tolerant/resistant species would be the same across communities (for example, if the disease is caused by a specialist parasite, there could be a whole range of tolerant/resistant species). Even if the tolerant/resistant species are the same across communities, the species impacted by disease could be more affected by stochastic processes as they experience symptoms or their populations decline. Note that I’m not asking you to put these specific alternatives in the text. My aim is to demonstrate why it’s not intuitive to me that this should be the expected outcome over other outcomes. The “system-specific hypothesis” is presented on L106-110 and hinted on L425-428: reducing disease with fungicide drives convergence of communities because more impacted species become more dominant. This hypothesis is subject to the same question as the first hypothesis.

Reviewer #2: (No Response)

7. PLOS authors have the option to publish the peer review history of their article (what does this mean?). If published, this will include your full peer review and any attached files.

Reviewer #1: No

Reviewer #2: No

---

## [Author Response · Author response to Decision Letter 1]

22 Sep 2023

Dear Editor,

Thank you for the opportunity to submit a revised version of our manuscript titled “Disease decreases variation in host community structure in an old-field grassland.” We thank the reviewers for their helpful comments and have revised the manuscript in response to their feedback. In response to reviewer 1, we have clarified parts of the introduction to better differentiate the hypotheses in the manuscript. We have provided a response letter to the comment from the reviewer with line references below in bold italics and a tracked changes version of our manuscript. 

We feel these changes have improved the quality of our manuscript, and we hope that you will now find this work suitable for PLOS ONE. Thank you for your efforts in the publication of our research.

Sincerely, 

Rita Grunberg

Review Comments to the Author

Reviewer #1: The authors successfully addressed almost all of my comments and I appreciated their responses. However, I still had to read the introduction a few times to understand the motivating theory/hypotheses. I know I already gave a lengthy comment on this in my last review, so I will try to explain it differently this time. I think that all the ideas are there but are difficult to understand due to unbalanced presentation of hypotheses and implicit assumptions. It also might help to reconsider the order of ideas in the introduction. The two potential outcomes of disease on convergence are presented in an unbalanced way because one is the main conceptual hypothesis and the other is system-specific hypothesis. The “main conceptual hypothesis” is presented on L23-24, L56-58, and L73-75: disease drives convergence of communities because species that are more tolerant/resistant become more dominant. This hypothesis needs more citations and/or explanation of implicit assumptions because it is unclear why the more tolerant/resistant species would be the same across communities (for example, if the disease is caused by a specialist parasite, there could be a whole range of tolerant/resistant species). Even if the tolerant/resistant species are the same across communities, the species impacted by disease could be more affected by stochastic processes as they experience symptoms or their populations decline. Note that I’m not asking you to put these specific alternatives in the text. My aim is to demonstrate why it’s not intuitive to me that this should be the expected outcome over other outcomes. The “system-specific hypothesis” is presented on L106-110 and hinted on L425-428: reducing disease with fungicide drives convergence of communities because more impacted species become more dominant. This hypothesis is subject to the same question as the first hypothesis.

Author: Thank you for the clarification. In the introduction, we have included more references in certain parts to support our hypotheses. In addition, we added text that describes three specific scenarios: convergence, spatial variation, and stochastic variation, to define our hypotheses and expectations more clearly. The description of the convergence scenario is on lines 57-59. The spatial variation scenario is on lines:63-65. The stochastic variation scenario is on lines 67-70. These scenarios are also referred to at the end of the introduction (lines: 111-123) and the discussion (lines: 443-445, 462-465, 572-274). We believe this addition will make the introduction clearer and help the reader understand our expectations. 

Reviewer #2: (No Response)

---

## [Decision Letter · Decision Letter 2]

16 Oct 2023

Disease decreases variation in host community structure in an old-field grassland

PONE-D-23-04650R2

Dear Dr. Grunberg,

We’re pleased to inform you that your manuscript has been judged scientifically suitable for publication and will be formally accepted for publication once it meets all outstanding technical requirements.

Kind regards,

Harald Auge

Academic Editor

PLOS ONE

Additional Editor Comments (optional):

Reviewers' comments:

Reviewer's Responses to Questions

**Comments to the Author**

1. If the authors have adequately addressed your comments raised in a previous round of review and you feel that this manuscript is now acceptable for publication, you may indicate that here to bypass the “Comments to the Author” section, enter your conflict of interest statement in the “Confidential to Editor” section, and submit your "Accept" recommendation.

Reviewer #1: All comments have been addressed

2. Is the manuscript technically sound, and do the data support the conclusions?

Reviewer #1: Yes

3. Has the statistical analysis been performed appropriately and rigorously? 

Reviewer #1: Yes

4. Have the authors made all data underlying the findings in their manuscript fully available?

Reviewer #1: Yes

5. Is the manuscript presented in an intelligible fashion and written in standard English?

Reviewer #1: Yes

6. Review Comments to the Author

Reviewer #1: (No Response)

7. PLOS authors have the option to publish the peer review history of their article (what does this mean?). If published, this will include your full peer review and any attached files.

Reviewer #1: No

---

## [Editor Report · Acceptance letter]

19 Oct 2023

PONE-D-23-04650R2 

Disease decreases variation in host community structure in an old-field grassland  

Dear Dr. Grunberg:

I'm pleased to inform you that your manuscript has been deemed suitable for publication in PLOS ONE. Congratulations! Your manuscript is now with our production department. 

Kind regards, 

on behalf of

Dr. Harald Auge 

Academic Editor

PLOS ONE